# Functionalized PLGA-Based Nanoparticles with Anti-HSV-2 Human Monoclonal Antibody: A Proof of Concept for Early Diagnosis and Targeted Therapy

**DOI:** 10.3390/pharmaceutics16091218

**Published:** 2024-09-18

**Authors:** Melinda Mariotti, Noah Giacon, Ettore Lo Cascio, Margherita Cacaci, Simona Picchietti, Maura Di Vito, Maurizio Sanguinetti, Alessandro Arcovito, Francesca Bugli

**Affiliations:** 1Dipartimento di Scienze Biotecnologiche di Base, Cliniche Intensivologiche e Perioperatorie, Università Cattolica del Sacro Cuore, Largo A. Gemelli 8, 00168 Rome, Italy; melinda.mariotti@unicatt.it (M.M.); noah.giacon@unicatt.it (N.G.); ettore.locascio@unicatt.it (E.L.C.); margherita.cacaci@unicatt.it (M.C.); maura.divito@unicatt.it (M.D.V.); maurizio.sanguinetti@unicatt.it (M.S.); 2Dipartimento di Scienze di Laboratorio e Infettivologiche, Fondazione Policlinico Universitario A. Gemelli IRCCS, 00168 Rome, Italy; 3Department for Innovation in Biological, Agro-Food and Forest Systems (DIBAF), University of Tuscia, Largo dell’Università snc, 01100 Viterbo, Italy; picchietti@unitus.it; 4Fondazione Policlinico Universitario “A. Gemelli”, IRCCS, Largo A. Gemelli 8, 00168 Roma, Italy

**Keywords:** nanoparticles, nano-immuno-probe, nanotechnology, Fab, PLGA

## Abstract

**Background:** Functionalized nanoparticles (NPs) represent a cutting edge in innovative clinical approaches, allowing for the delivery of selected compounds with higher specificity in a wider time frame. They also hold promise for novel theranostic applications that integrate both diagnostic and therapeutic functions. Pathogens are continuously evolving to try to escape the strategies designed to treat them. **Objectives:** In this work, we describe the development of a biotechnological device, Nano-Immuno-Probes (NIPs), for early detection and infections treatment. Human Herpes Simplex Virus 2 was chosen as model pathogen. **Methods:** NIPs consist of PLGA-PEG-Sulfone polymeric NPs conjugated to recombinant Fab antibody fragments targeting the viral glycoprotein G2. NIPs synthesis involved multiple steps and was validated through several techniques. **Results:** DLS analysis indicated an expected size increase with a good polydispersity index. Z-average and z-potential values were measured for PLGA-PEG-Bis-Sulfone NPs (86.6 ± 10.9 nm; –0.7 ± 0.3 mV) and NIPs (151 ± 10.4 nm; −5.1 ± 1.9 mV). SPR assays confirmed NIPs’ specificity for the glycoprotein G2, with an apparent *K_D_* of 1.03 ± 0.61 µM. NIPs exhibited no cytotoxic effects on VERO cells at 24 and 48 h. **Conclusions:** This in vitro study showed that NIPs effectively target HSV-2, suggesting the potential use of these nanodevices to deliver both contrast agents as well as therapeutic compounds.

## 1. Introduction

In recent years, nanotechnology has gone through remarkable developments, emerging as significant in several applications, including the food industry, agriculture, and cosmeceuticals [1,2,3]. Moreover, it offers new solutions to overcome the limitations of conventional medicine, with exceptional progress in cancer and diabetes treatment [4,5,6,7,8], regenerative medicine, ocular therapy [9], tissue imaging [10,11], vaccines [12,13], and infections [14]. In particular, nanoparticles (NPs) have been intensively studied for both diagnosis and therapy. As diagnostic tools, NPs equipped with signal molecules are considered an emerging class of contrast imaging agents, showing multimodal signal and multiplexing capabilities, which allow them to be detected by several imaging methods and to detect different molecular targets, respectively [15,16]. For therapeutic applications, NPs represent ideal drug delivery systems, ensuring targeted delivery and controlled release of drugs at specific sites and within defined time windows, providing an alternative to common therapies. With traditional administration methods, drugs are extensively distributed at the systemic level, and high and repeated doses are often required to achieve therapeutically effective concentrations [17,18]. However, loading a drug into an NP increases its stability, prolongs its circulation lifetime, and minimizes therapy-related toxicity [19]. Among the several strategies that have been developed to promote NPs accumulation in a site of interest, active targeting approaches are based on the conjugation of ligands to the NPs surface and depend on the molecular recognition between the ligand-functionalized NP and its specific target expressed at the site of interest [20,21]. Different moieties have been evaluated as ligands for the fabrication of targeted NPs: small molecules like folic acid [22], carbohydrates [23,24,25], aptamers [26], peptides [27], and especially antibodies [28]. Thanks to their high affinity and specificity to a target antigen, antibodies are among the most successfully employed ligands. However, they are large Y-shaped proteins, constituted of two identical light and heavy chains held together by disulphide bonds, and it could be difficult to produce them as recombinant molecules. Thus, smaller antibody fragments (Fabs) are often preferred for antibody-based NPs, since they maintain the specificity of the whole protein without triggering the complement activation [29,30,31]. Furthermore, NPs are intensively studied for their potential to combine diagnostic and therapeutic functions within a single entity, thus enabling simultaneous disease diagnosis and treatment, the selection of optimal treatment, and the monitoring of therapeutic efficacy over time [19,32,33]. The novelty and significance of this study lie in the development of a new biotechnological device, called Nano-Immuno-Probe (NIP). This device consists of PLGA-PEG-Bis-Sulfone polymeric nanoparticles conjugated with recombinant Fab antibody fragments that specifically target the glycoprotein G2 (gG2) of Human Herpes Simplex Virus 2 (HSV-2). PLGA (Poly(lactic-co-glycolic acid)) and PEG (Polyethylene glycol) are widely used polymers in nanotechnology due to their biocompatibility, biodegradability, and ability to evade the immune system [34]. The incorporation of bis-sulfone, a benzoic acid derivative, enables the functionalization of these nanoparticles with His-tagged or thiolated compounds, such as peptides, protein domains, or antibodies [35]. This results in nanoparticles that are highly stable, exhibit reduced immunogenicity, and provide efficient drug delivery and precise targeting capabilities [36]. HSV-2 was chosen as the model because a human monoclonal anti-HSV-2 antibody had previously been obtained from a phage display combinatorial library constructed from the iliac crest bone marrow of an infected immunocompromised patient; furthermore, the Fab had already been characterized for diagnostic purposes and targets the glycoprotein G2 [37].

In this work, NIPs were fabricated by a multi-step process that involved synthesizing PLGA-PEG-NH_2_, conjugating Bis-Sulfone to it, and then eliminating Toluene-sulphonic acid to obtain PLGA-PEG-Mono-Sulfone NPs. The final polymer of the reaction, PLGA-PEG-Bis-Sulfone, was used to synthesize nanoparticles via the nanoprecipitation technique. The resulting colloids were then functionalized with the antibody, creating the NIPs. These devices can deliver both contrast agents and therapeutic compounds, a relatively novel approach in personalized medicine. This dual capability enables a combined diagnostic and therapeutic strategy, potentially enhancing the efficiency and effectiveness of treatments while minimizing adverse effects.

To determine the correct synthesis and formation of NIPs, they were characterized using dynamic light scattering (DLS), nanoparticle tracking analysis (NTA), and transmission electron microscopy (TEM). Additionally, an anti-Fab antibody labelled with gold particles was used to confirm the expression of the targeting ligand on the surface of the NIPs; SPR enabled the assessment of the interaction between the NIPs and glycoprotein G2. Finally, the cytotoxicity profile of the NIPs was evaluated on VERO cells.

## 2. Materials and Methods

### 2.1. Escherichia coli Cell Strain Used for Cloning and Expressing Recombinant Anti-HSV-2 Fab

The *Escherichia coli* strain XL-1 Blue (RBC Bioscience, New Taipei City, Taiwan) was used as a host cell in the subcloning and expression experiments. The recombinant plasmid was transformed into *E. coli* cells using standard methods. Following transformation, cells were grown on Luria Bertani LB agar plates (Condalab, Madrid, Spain) and inoculated in Super Broth SB (3.5% tryptone, 2% yeast extract, 0.5% NaCl, 1N NaOH), both containing the antibiotics ampicillin (Amp, 100 µg/mL) and tetracycline (Tet, 20 µg/mL). Single antibiotic-resistant recombinant colonies were selected for protein expression.

### 2.2. Construction of pComb3/TIG Vector Encoding Recombinant Histidine Tagged Anti-HSV-2 Fab

The human monoclonal anti-HSV-2 antibody was previously obtained from a phage display combinatorial library constructed from iliac crest bone marrow of infected immunocompromised patients and characterized for diagnostic purposes by Bugli et al. [37]. Multiple subcloning procedures were performed to add a Histidine tag to the C-terminal end of the anti-HSV-2 Fab Heavy Chain (HC), necessary for the chemical conjugation to NPs.

### 2.3. Light and Heavy Chains Subcloning

The light and heavy chains were subcloned into pComb3/TIG to obtain the pComb3/TIG-LC-HC_HIS_, which encodes the Histidine tagged anti-HSV-2 Fab. For the details of the cloning procedure, see Appendix A.

### 2.4. Sequencing

The anti-HSV-2 Fab heavy and light chains were sequenced using the pComb3/TIG-LC-HC_HIS_ without CP3 plasmid as a template. In every Sanger fluorescence-based sequencing reaction, the BigDye terminator v3.1 and its 5× Buffer (Applied Biosystems by Thermo Fisher Scientific, Waltham, MA, USA) were used. The nucleotide sequence of primers is shown in Appendix A of Appendix A. The sequencing reactions were purified with the DyeEx 2.0 Spin Kit (Qiagen, Hilden, Germany) to avoid the interference of salts, unincorporated dye terminators, and dNTPs; each reaction was suspended in formamide and the automatic sequencer 3130 Genetic Analyzer (Applied Biosystems by Thermo Fisher Scientific, Waltham, MA, USA) was used. The data were analyzed with the Chromas Pro 6.0 software.

### 2.5. Expression and Purification of the Recombinant Anti-HSV-2 Fab

*E. coli* XL-1 Blue transformants harboring pComb3/TIG-LC-HC_HIS_ without CP3 were inoculated in 10 mL of sterilized SB medium with 100 µg/mL Amp and 20 µg/mL Tet, and cultivated overnight at 37 °C. This starter culture was used to sub-inoculate (1:50) 1 L of SB medium and cultivated at 37 °C with vigorous shaking (220 rpm), until reaching the OD_600_ of 0.8. IPTG (isopropyl β-D1-thiogalactopyranoside) was added to a final concentration of 1 mM, 50 µg/mL Amp was added again to reconstitute, and the culture continued at 30 °C overnight. Cells were harvested by centrifugation at 3000 rpm at 4 °C for 20 min, the culture medium was discarded, and cells washed in PBS. Cells were pelleted by centrifugation at 5000 rpm at 4 °C for 30 min and resuspended in 100 mL of PBS. Chicken Egg White Lysozyme Solution (Merck Millipore Ltd., Burlington, MA, USA) was added to a final concentration of 100 µg/mL, and cells were incubated on ice for 30 min. The samples were sonicated eight times for 60 s, and Pierce Protease Inhibitors (Thermo Fisher Scientific, Waltham, MA, USA) were added after the first sonication; the bacterial lysates were centrifuged at 18,000 rpm at 4 °C for 45 min, and the supernatant recovered and filtered at 0.22 µm. The Fab was purified on a chromatographic column using a homemade Protein G-Human Fab resin and later analyzed as described in Appendix A.

### 2.6. Surface Plasmon Resonance

The interactions between the commercial recombinant gG2 glycoprotein (ligand) and the His-tagged anti-HSV-2 Fab or the functionalized NIPs (analytes), were measured using the Surface Plasmon Resonance (SPR) technique using a Biacore X100 instrument (Biacore, Uppsala, Sweden). The gG2 ligand was immobilized on a Sensor Chip CM7 (Biacore AB, Uppsala, Sweden) at 50 µg/mL. The immobilization was obtained via amine coupling (with EDC/NHS solutions) in accordance with the instructions of the manufacturer. In this process, the carboxyl groups on the CM7 chip are first activated with EDC (1-ethyl-3-(3-dimethylaminopropyl) carbodiimide) and NHS (N-hydroxysuccinimide). The protein is then introduced to the activated chip, allowing the amine groups to form a covalent bond with the chip. After immobilization, any remaining reactive NHS esters on the chip surface are deactivated using a solution of ethanolamine. This prevents nonspecific binding during the SPR analysis. Following the immobilization process, affinity analysis is performed. PBS 1X (containing 10 mM phosphate buffer pH 7.4, 0.137 M NaCl and 2.5 mM KCl) was filtrated and used as a running buffer and the binding experiments were performed with a flow rate of 30 µL/min at 25 °C; the association phase was monitored for 180 s, while dissociation was monitored for 300 s. The concentrations analyzed in the SPR assay for both the analytes refer to the concentration of the Fab species and were obtained by successive dilutions, halving the concentration in each step starting from 4 µM (i.e., 200 µg/mL) for the Fab alone and 1 µM of Fab equivalent for the NIPs. Each experiment was carried out using a minimum of five different analyte concentrations, and to verify the reproducibility of data, at least one concentration was repeated in duplicate. The regeneration of the chip surface was achieved by the addition of 2 M NaCl for 30 s before the start of each new cycle. Subsequently, regeneration was performed using a Glycine-HCl buffer (0.1 M, pH 2.5) at the end of the analysis. When the experimental data met the quality criteria, data were analyzed using the Biacore X100 Evaluation Software 2.0.1 plus package. An affinity steady state model was applied to fit the data, as kinetic parameters were out of the range measured by the instrument, but an equilibrium signal of interaction was clearly detected. Therefore, a specific *K_D_* was determined with a confidence interval associated with a standard error value to avoid any bias.

### 2.7. Nano-Immuno-Probes Synthesis

Nanoparticles were synthesized according to the method described in the following article [35]. A summary diagram of the various steps required to synthesize NIPs is shown in Figure 1a. Briefly, 0.97 g of commercial PLGA-COOH (MW 10 kDa, 0.1 mmol) (Nanosoft polymers, Winston-Salem, NC, USA) was activated and converted to PLGA-NHS with an excess of N,N′-Dicyclo-hexylcarbodiimide (DCC) and N-Hydroxysuccinimide (NHS) (both from Fluorochem Ltd., Glossop, UK). Quickly, PLGA-COOH was dissolved in 10 mL of dichloromethane (DCM) (Merck, Darmstadt, Germany) followed by the addition of 2.5 equivalents of DCC and NHS. The reaction was left under magnetic stirring for 20 h at room temperature. Once activated, insoluble dicyclohexyl urea was filtered and the final product was dried under vacuum to be conjugated to PEG. PLGA-NHS (900 g) were dissolved in 10 mL of DCM before the addiction of 540 mg of NH_2_-PEG-NH_2_ (MW 3 kDa, 0.18 mmol) (Sigma-Aldrich, Saint Louis, MO, USA). The reaction was left under magnetic stirring overnight at room temperature and the resultant polymer was triturated with methanol (Fluka Chemicals, Buchs, Switzerland), to be later dried under vacuum. Bis-Sulfone activation was achieved by dissolving 100 mg of Bis-Sulfone (0.16 mmol) (Fluorochem Ltd., Glossop, UK) in 10 mL of DCM. Both DCC and NHS were added in a stoichiometric excess of two times compared to Bis-Sulfone. The reaction was left under gentle stirring for 3 h at room temperature and the resulting product was filtered, triturated with diethyl ether (Merck, Darmstadt, Germany), and lastly dried under vacuum. After the Bis-Sulfone activation, 500 mg of PLGA-PEG-NH_2_ (0.038 mmol) were dissolved in 10 mL of DCM. Once solubilized, 1.1 equivalents of activated Bis-Sulfone were added to the solution. The reaction was allowed to proceed under magnetic stirring overnight. The resulting PLGA-PEG-Bis-Sulfone polymer was dried under vacuum and used to prepare the nanoparticles. To obtain PLGA-PEG-Bis-Sulfone NPs, the nanoprecipitation method (see Figure 1b) was performed: 100 mg of PLGA-PEG-Bis-Sulfone were dissolved in 1 mL di tetrahydrofuran (Applied Biosystems by Thermo Fisher Scientific, Waltham, MA, USA); then, 200 µL of the resultant product were added dropwise to 5 mL of stirring water. The reaction was left to proceed overnight, and the next day PLGA-PEG-Bis-Sulfone NPs were analyzed through DLS.

The conversion of PEG-Bis-Sulfone to PEG-Mono-Sulfone was induced through the addition of an appropriate buffer to the NPs solution. Briefly, 150 µL of a solution constituted of 100 mM NaCl, 20 mM EDTA (Sigma-Aldrich, Saint Louis, MO, USA), and 500 mM phosphate buffer pH 8 (Carlo Erba, Val de Reuil, France) was added to 1.35 mL of PLGA-PEG-Bis-Sulfone NPs. The reaction was incubated at 37 °C for ~6 h. 500 µL of the His tagged anti-HSV-2 dialyzed Fab (4 µM, in PBS 1×) was added to 1.5 mL of PLGA-PEG-Mono-Sulfone NPs for the NPs functionalization reaction and incubated with gentle agitation at room temperature overnight. To avoid unspecific signals, the unbound Fab that did not react with the NPs Mono-Sulfone was removed from the solution using the VivaSpin 6 mL Concentrator, 100,000 MWCO (VivaScience by Sartorius, Göttingen, Germany). The sample was loaded into the VivaSpin Concentrator, centrifuged at 1500 rpm for 2.5 min, and 1 mL of the flow through (referred to as unbound Fab) was collected for further analysis. After five washes in PBS, the NIPs without the unbound Fab were recovered in 1 mL of PBS.

### 2.8. Dynamic Light Scattering

DLS experiments were performed using the Zetasizer Nano S (Malvern Instruments, Malvern, UK) equipped with a 4 mW He-Ne laser (633 nm). Measurements were carried out at 25 °C at an angle of 173° from the incident beam. The Z-Average diameters of the scattering particles were calculated by peak-intensity and number analysis. The Zeta-Potential values were collected using DTS1070 (Disposable Folded Capillary cell). Samples were diluted 1:100 with distilled water before the analysis.

### 2.9. Transmission Electron Microscopy

Droplets of NPs and NIPs suspensions (10 μL) were placed on formvar-carbon coated grids and allowed to adsorb for few min. The adsorbed samples were processed for negative staining by washing the specimen grid on a drop of negative stain solution (2% uranyl acetate dissolved in distilled water) and then repeating this step once more leaving the specimen grid on a new drop of negative stain solution for 120 s. Contrast agents were used only for low-contrast materials (e.g., PLGA). Samples were observed with a JEOL 1200 EX II electron microscope (JEOL Ltd., Tokyo, Japan). Micrographs were acquired with an Olympus SIS VELETA CCD camera (Shinjuku City, Tokyo, Japan) equipped with the iTEM software 2009.

### 2.10. Immunoelectron Microscopy

For immunogold staining (IGS), NPs and NIPs suspensions were adsorbed on formvar-carbon coated grids, as described in the previous paragraph. Non-specific antigens were blocked with 0.5% Bovine Serum Albumin (BSA) in PBS (pH 7.4) for 15 min. Subsequently, samples were incubated for 60 min in a moist chamber with a polyclonal antibody anti-human Fab conjugated to 25 nm gold particles (St John’s Laboratory Ltd., London, UK), and diluted 1:500 in 0.1% Tween20 and 1% BSA in PBS (pH 8.2). After rinsing in 0.5% BSA in PBS and then in PBS (5 min each), the grids were washed three times with distilled water (5 min each). PBS was substituted with the anti-human Fab antibody in the negative controls. Samples were subsequently stained with uranyl acetate and observed with a JEOL JEM EX II transmission electron microscope (JEOL Ltd., Tokyo, Japan) at 100 kV. Micrographs were acquired with an Olympus SIS VELETA CCD camera equipped with the iTEM software 2009.

### 2.11. Cytotoxicity Test

VERO eukaryotic cells were cultured at 37 °C in a humidified environment (CO_2_ 5%) in MEM containing L-glutamine, supplemented with 10% Fetal Bovine Serum (FBS) and 1% Penicillin-Streptomycin Antibiotic (all Gibco by Thermo Fisher Scientific, Waltham, MA, USA). A total of 50,000 cells/well in basal medium were seeded into a 96-well plate (Corning Incorporated, Kennebunk, MA, USA) until a sub-confluent monolayer was achieved. Cells were treated with different concentrations of NIPs to achieve the following NIPs final percentages: 50%, 25%, 12.5%, and 6.25% *v*/*v*. Untreated cells were used as the control. After 24 and 48 h of incubation, cellular viability was evaluated by the MTS assay, using the MTS Cell Proliferation Assay Kit (Abcam Plc, Cambridge, UK) according to the manufacturer instructions. The optical density (OD) of the solution in each well was determined with a plate reader at a wavelength of 490 nm. Cell vitality was calculated according to the following equation: Viability (%): (OD sample/OD control) × 100.

### 2.12. Statistical Analysis

Statistical analysis was performed using GraphPad Prism version 9.1.2 for Windows, GraphPad Software (San Diego, CA, USA).

## 3. Results

### 3.1. Cloning Strategy

From the anti-HSV-2 original plasmid, in which the genes encoding the monoclonal antibody light and heavy chains had been previously cloned, the LC gene was subcloned into the pComb3/TIG vector. Subsequently, the PCR amplifications of the HC gene with the addition of a four Histidine tag at the C-terminal end was accomplished. The amplifications performed with the specific primer for the His tag addition showed no difference compared to those with the unmodified primer (Figure 2). The HC was cloned into the LC containing vector, resulting in the anti-HSV-2 pComb3/TIG-LC-HC_HIS_ vector verified by restriction analysis; positive clones were identified.

### 3.2. Determination of the Amino Acid Sequence of the Fab

The amino acid sequence of the HC_HIS_ and LC variable regions of the anti-HSV-2 Fab was inferred from their DNA sequence. Seven domains were sequenced for both the HC_HIS_ and LC of 122 and 105 amino acids (Appendix A), respectively. The sequencing of the CH1 constant region of the heavy chain showed four Histidines at the C-terminal end.

### 3.3. Expression and Purification of Recombinant Anti-HSV-2 Fab

Small-scale optimization experiments were conducted to characterize the best expression conditions. The Fab fragment was purified by immunoaffinity chromatography on a human anti-Fab sepharose column starting from 1 L of induced cell culture. The eluted fractions were analyzed on polyacrylamide gel under reducing conditions; a band of about 25 kDa was detected as expected. Elution fractions 2 and 3 with the highest Fab concentration were pooled and dialyzed against PBS. The pooled elution fractions were analyzed by SDS-PAGE before and after the dialysis. As can be seen in Figure 3, dialysis did not alter the Fab concentration. The final yield obtained was measured at 200 µg/mL (4 µM).

### 3.4. NIPs Synthesis and Characterization

NIPs were fabricated according to the procedures described in Methods (paragraph 2.7). Briefly, the PLGA-PEG-NH_2_ co-polymer was synthesized from PLGA-NHS and NH_2_-PEG-NH_2_, the Bis-Sulfone was conjugated to it, and the PLGA-PEG-Bis-Sulfone NPs were achieved by nanoprecipitation. Through the elimination of the Toluene-sulphonic acid at basic pH, the PLGA-PEG-Mono-Sulfone NPs were obtained, and the reaction with the C-terminus His tagged anti-HSV-2 Fab resulted in the NIPs. To verify the synthesis process, several analyses were performed on the intermediate and final products. Table 1 shows the z-average diameters and polydispersity indexes of PLGA-PEG-NH_2_ NPs and PLGA-PEG-Bis-Sulfone NPs and NIPs, measured by DLS. Nanoparticle tracking analysis on PLGA-PEG-Bis-Sulfone NPs provided a high-resolution particle size distribution profile from 90 to 200 nm and a concentration measurement of 10^10^ particles/mL (see Appendix A in the Appendix A).

Dynamic Light Scattering (DLS) was employed to characterize the Zeta Potential (Z-potential) of nanoparticles both before and after functionalization. The final results are presented in Table 2. To ensure significance, three independent measurements were conducted for each sample. The Zeta Potential value of nanoparticles alone was determined to be −0.7 ± 0.3 mV, whereas the Zeta Potential of NIPs was found to be −5.1 ± 1.9 mV. The lower Z-potential of NIPs can be attributed to the introduction of negatively charged groups from the Fab fragment causing an alteration in surface charge distribution and accessibility.

### 3.5. NIPs Ultrastructural Characterization and Localization of the Fab Ligand on NIPs

Transmission electron microscopy (TEM) was used to obtain a direct visualization of the nanoparticles with high resolution. When TEM is applied to visualize nanoparticles, a treatment of negative staining is necessary to describe the ultrastructure and potential alterations. As shown in Figure 4a,b, NPs and NIPs had a spherical shape and size heterogeneity. No changes in nanoparticle conformation, alteration of membrane curvature, or formation of roughening or surface ruptures were observed in NIPs compared to NPs. Moreover, an anti-Fab antibody labelled with gold particles was used in order to recognize the available epitopes of the anti-HSV-2 Fab, and used as a specific ligand to functionalize the PLGA-PEG-Mono-Sulfone NPs. Figure 4c shows a magnification depicting this interaction. Gold particles were found on the surface of NIPs, revealing the expression of the targeting ligand. As expected, the staining was not observed on the surface of NPs or in negative controls (Figure 4d).

### 3.6. Validation of Specificity of NIPs

To avoid an unspecific signal, at the end of the NIPs synthesis process, the Fab not conjugated to the NPs surface was removed from the solution using twin vertical PES membranes with a molecular weight cut off (MWCO) of 100 kDa. The removal of the unbound Fab was verified through a Western blot performed on the solution that flowed through the PES membranes (FT), and on the free anti-HSV-2 Fab. The free Fab was analyzed at the same concentration used in the binding reaction with the NPs. As shown in Figure 5, a slight band corresponding to the Fab fragment size (~25 kDa) was visible in the FT, much less intense than the band corresponding to the free Fab. The unbound Fab signal was so imperceptible that it was assumed that all of the anti-HSV-2 Fab fragment used in the reaction was bound to the NPs. For this reason, the concentration of the Fab bound to the NPs surface of NIPs was estimated at 50 µg/mL (1 µM). The anti-HSV-2 Fab fragment had already been characterized and it targeted the HSV-2 glycoprotein G2; once the unbound Fab had been removed, the NIPs ability to specifically recognize and bind to the gG2 was investigated through SPR, following the procedures described in materials and methods. In particular, a commercial recombinant gG2 was immobilized on a CM7 sensor chip and used as a ligand in the assay, whereas the NIPs were used as an analyte. The optimal experimental setup was settled, and the analyte was injected at five different concentrations using a multi-kinetic mode. As shown by the scatchard plot in Figure 6a, a major concentration of NIPs was related to an increase of the Response Unit (RU); an apparent *K_D_* of 1.03 ± 0.61 µM was estimated (Table 3). SPR was performed to assess the interaction between the free anti-HSV-2 Fab and the gG2 (Figure 6b) as well. Table 3 shows the estimated *K_D_* value obtained through a kinetic analysis. No interaction was measured between the NPs and the gG2.

### 3.7. NIPs Cytotoxicity

The NIPs cytotoxicity profile was assessed on VERO cells through the MTS assay. As shown in Figure 7, after 24 and 48 h, no cytotoxic effect was observed in the presence of NIPs at all of the four tested concentrations (50%, 25%, 12.5%, and 6.25% *v*/*v*). According to ISO 10993-5, a reduction of cell viability by >30% should be considered a cytotoxic effect [38]: as can be seen in Figure 7, all the median viabilities were indeed above 70%. No cytotoxic effect was observed on cells in the presence of PLGA-PEG-Bis-Sulfone NPs at a concentration equal to 50% *v*/*v*, at 24 and 48 h.

## 4. Discussion

Nanotechnology has become pivotal in various fields, particularly in overcoming medical challenges [6,7,8,9,10,11,12,13,14,39]. This study addresses the urgent need for highly sensitive diagnostic and therapeutic solutions, focusing on developing a biotechnological device intended for early detection of specific pathogens with a great clinical impact and, eventually, for the direct in situ treatment of the infection. HSV-2, chosen as the model for this project, is a common virus of the *Herpesviridae* family. Primarily known for causing genital herpes, HSV-2 is a neurotropic virus with significant clinical impact that poses challenges due to its ability to establish latent infections and reactivate periodically. Despite its global prevalence and impact on quality of life, antivirals are the current standard medication to prevent viral reactivation [40,41,42,43]. The human monoclonal anti-HSV-2 Fab, obtained from an immunocompromised patient’s bone marrow, targets glycoprotein G2 [37]. To develop NIPs, PLGA-PEG-Bis-Sulfone NPs needed Fabs functionalization, involving subcloning procedures to add a four Histidine tag to the antibody. The LC gene was cloned into the pComb3/TIG vector, and PCR amplifications added a four Histidine tag to the HC gene using a modified constant region primer (CG1z_4HIS_). The CG1z_4HIS_ primer showed good quality amplifications and no differences compared to those with the unmodified one (CG1z), and unlike the CG1z_6HIS_ primer, which would have added six Histidines. After subcloning HC_HIS_ into the LC containing vector, the monoclonal anti-HSV-2 Fab was expressed in *E. coli* XL-1 Blue cells, assembling in the periplasmic space via disulphide bond formation. Fab extraction was achieved through mild ultrasound cell disruption, followed by successful purification using immunoaffinity chromatography. In this study, NIPs were created using a method outlined in Section 2. Various types of NPs with specific features can be designed as needed. Poly Lactic-co-Glycolic Acid (PLGA) was selected for its Food and Drug Administration (FDA) and European Medicines Agency (EMA) approval, tunable mechanical properties, biocompatibility, and biodegradability [44]. PLGA can be polymerized with poly(ethylene glycol) (PEG), enhancing nano drug delivery systems and increasing circulation half-life [45]. PLGA-PEG NPs have been employed for various medical applications, including cancer and Alzheimer’s disease treatment, inflammation management, and drug delivery [46,47,48,49,50,51]. To validate NIPs synthesis, several techniques were employed. DLS analysis indicated an expected size increase with good polydispersity index and correlation coefficient values. Z-averages of 86.6 ± 10.9 nm with a Zeta Potential value of –0.7 ± 0.3 mV for PLGA-PEG-Bis-Sulfone NPs and 151 ± 10.4 nm with a Zeta Potential of −5.1 ± 1.9 mV for NIPs were obtained, corroborated by nanoparticle tracking analysis showing a size distribution profile of 90 to 200 nm. Size is crucial for NP function, with smaller systems (<200 nm) facilitating tissue acceptance and micro-capillary crossing [19,52]. In an SPR assay, NIPs’ specificity for glycoprotein G2 was confirmed, showing no alteration in Fab stability or unfolding upon conjugation to NPs. However, the binding affinity of NIPs-gG2 was lower than that of the free anti-HSV-2 Fab. These results may be due to the low number of Fab fragments that functionalized the NIPs. Furthermore, as Fabs on NIPs are chemically bound to the NPs surface and not free in solution, their engagement with gG2 immobilized on the CM7 sensor chip in the SPR assay was limited. In fact, the unbound Fabs on NIPs may not contribute to increasing the binding affinity, even if they are indeed part of those NIPs. These data suggested that the interaction between NIPs and gG2 differs from the one evaluated for the free Fab, providing further evidence of the successful NIPs synthesis reaction. Finally, the data obtained from the toxicity test on VERO cells indicate that NIPs showed no cytotoxic effects at 24 and 48 h.

## 5. Conclusions

The in vitro experiments conducted in this study showed that Nano-Immuno-Probes successfully recognized and bound to the specific target of the anti-HSV-2 Fab, which was tethered to NPs surface. This suggests that NIPs could serve as effective tools for detecting specific pathogens and early identification of the infection sites, potentially enabling direct in situ treatment and significantly improving patients’ prognosis. NIPs can be produced through a fast synthesis with high yield and mild reaction conditions, making them a versatile and effective platform. However, scaling up production to industrial levels poses challenges in maintaining consistency and quality, and high costs.

Successful development of these nanodevices could greatly expand the pharmaceutical market for advanced drug delivery systems, with potential applications extending beyond viral infections to various diseases, thus enhancing treatment efficacy and personalized medicine approaches. A SWOT analysis attempt has been made and added to the Appendix A.

## Figures and Tables

**Figure 1 pharmaceutics-16-01218-f001:**
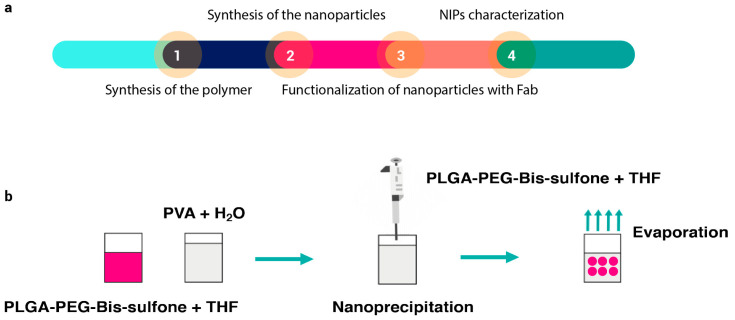
Panel (**a**) illustrates the sequence of steps involved in synthesizing and characterizing NIPs. The process begins with the synthesis of the polymer (Step 1), followed by the synthesis of nanoparticles via nanoprecipitation (Step 2). Once formed, the nanoparticles are functionalized with the antibody (Step 3), and finally, the particles undergo characterization (Step 4). Panel (**b**) depicts a schematic of the nanoprecipitation technique. In this technique, PLGA-PEG-Bis-sulfone polymer is dissolved in THF (tetrahydrofuran). This solution is then added dropwise to continuously stirred water. The mixture is left stirring until nanoparticles are formed.

**Figure 2 pharmaceutics-16-01218-f002:**
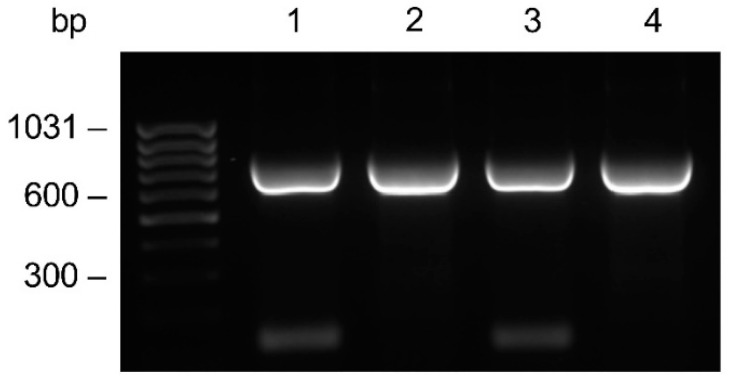
Comparative evaluation on the agarose gel of the HC gene amplification. Amplicons were generated by PCR using the CG1z_4HIS_ primer for the Histidine tag addition (lanes 1–2), or the CG1z unmodified primer (lanes 3–4) in combination with a mix of A (1–3) or F (2–4) variable primers. The anti-HSV-2 original plasmid was used as a template.

**Figure 3 pharmaceutics-16-01218-f003:**
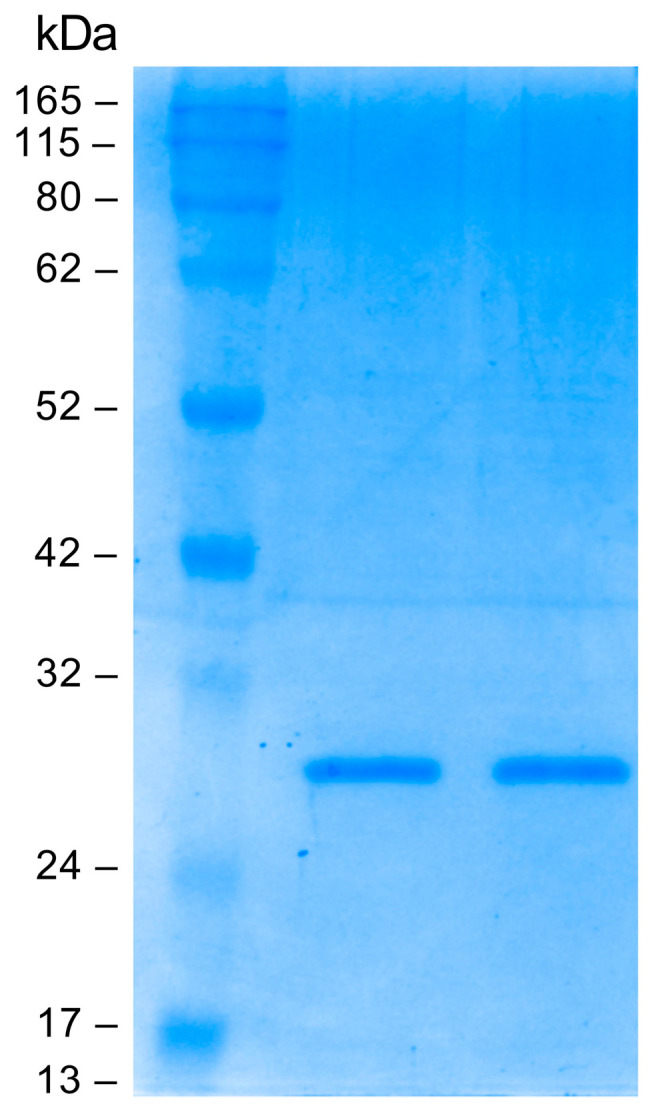
Sodium dodecyl sulfate polyacrylamide gel electrophoresis (SDS-PAGE; Coomassie blue staining) analysis: evaluation of dialysis efficiency. From left to right: protein molecular mass marker, anti-HSV-2 Fab pre-dialysis, anti-HSV-2 Fab post dialysis.

**Figure 4 pharmaceutics-16-01218-f004:**
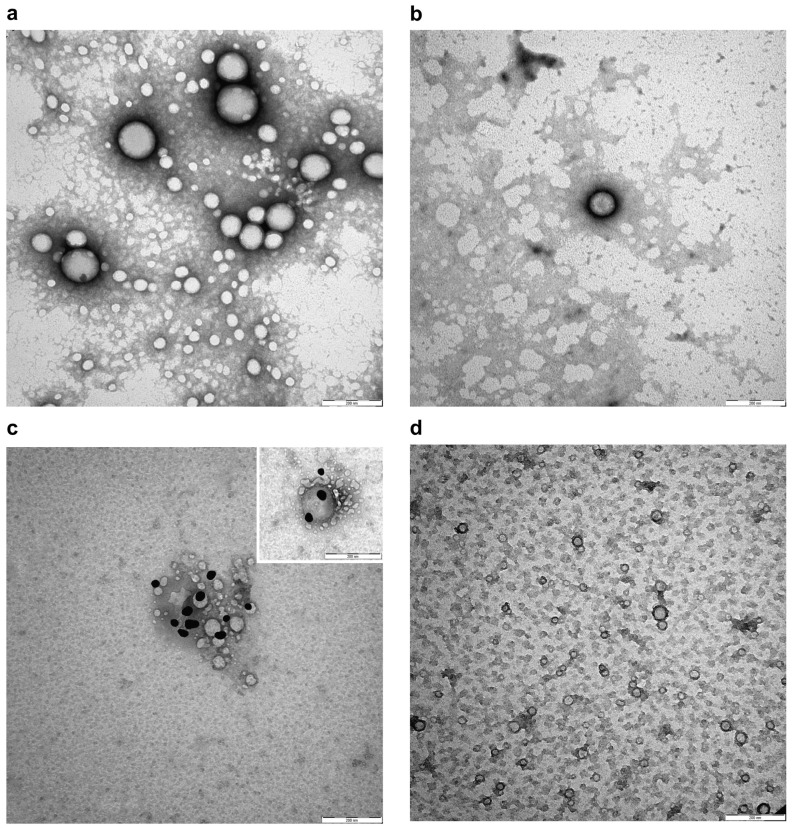
Negative staining (**a**,**b**) and representative immunogold images (**c**,**d**) of NPs and NIPs. (**a**) NPs showing spherical shape, size heterogeneity and smooth membranes; (**b**) Smooth membrane of spherical NIPs; (**c**) Detection of Fab epitopes on the surface of NIPs using gold conjugated Fab-specific antibody (dark particles); (**d**) NPs without staining on their surface in negative control. Bars: (**a**–**d**) 200 nm. Magnification rate: 100 K; scale bar: 200 nm.

**Figure 5 pharmaceutics-16-01218-f005:**
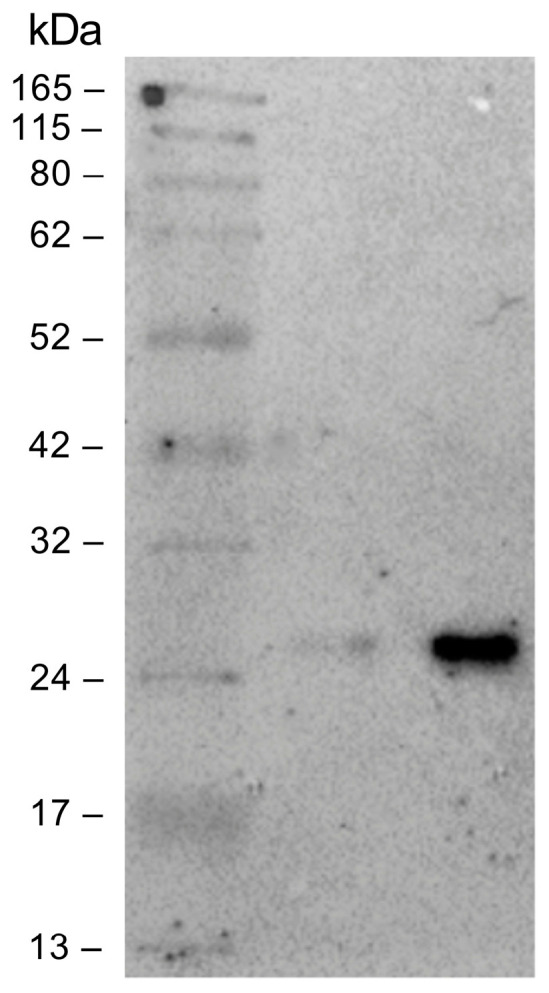
Immunoblot of the PES membranes that flowed through the solution (FT) and the free anti-HSV-2 Fab. The samples were separated by SDS-PAGE under reducing conditions and then transferred to the nitrocellulose membrane. From left to right: marker, unbound Fab, free anti-HSV-2 Fab (C: 1 µM).

**Figure 6 pharmaceutics-16-01218-f006:**
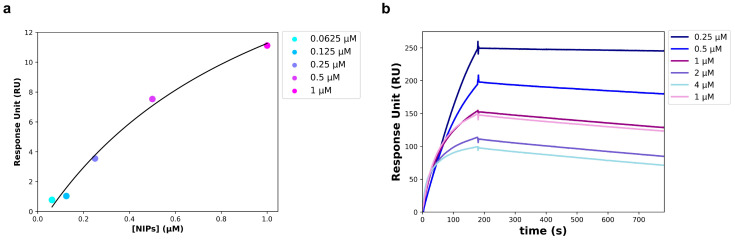
SPR analysis carried out on a CM7 sensor chip. (**a**) Scatchard Plot of the interaction between recombinant gG2 (ligand) and the NIPs (analyte). Data points were acquired starting from the 1:1 NIPs concentration and obtaining the others by successive 1:2 dilutions. (**b**) Sensogram of the interaction between recombinant gG2 (ligand) and the anti-HSV-2 Fab (analyte); data points were obtained at the following concentrations of the Fab: 4, 2, 1, 0.5 and 0.25 µM.

**Figure 7 pharmaceutics-16-01218-f007:**
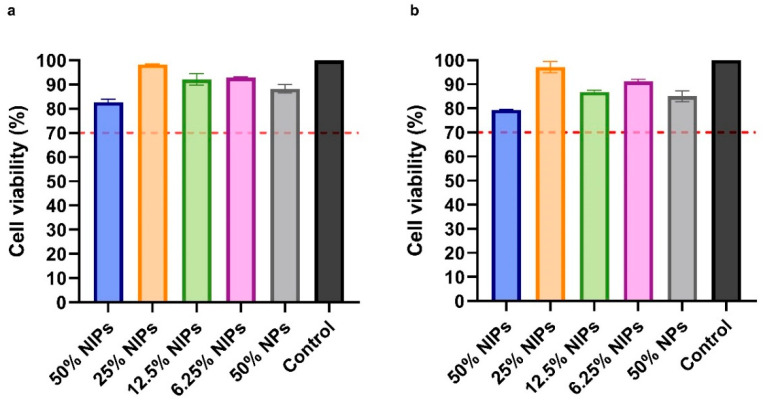
Cytotoxicity analysis with MTS assay on VERO cells at 24 (**a**) and 48 h (**b**). Cells were treated with four different NIPs concentrations: 50%, 25%, 12.5%, and 6.25% *v*/*v*. NPs were tested as well. The dotted red line at 70% indicates the in vitro cytotoxicity threshold.

**Table 1 pharmaceutics-16-01218-t001:** Z-average diameter and polydispersity index measured through DLS analysis.

Sample	z-Average (nm)	Polydispersity Index
PLGA-PEG-NH_2_ NPs	52.4 ± 4.46	0.181
PLGA-PEG-Bis-Sulfone NPs	86.6 ± 10.9	0.184
NIPs	151 ± 10.4	0.124

**Table 2 pharmaceutics-16-01218-t002:** Zeta potential values obtained through DLS analysis.

Sample	Zeta Potential (mV)
PLGA-PEG-Bis-Sulfone NPs	–0.7 ± 0.3 mV
NIPs	–5.1 ± 1.9 mV

**Table 3 pharmaceutics-16-01218-t003:** Parameters obtained by the SPR measurements.

Ligand	Analyte	*K_D_*
gG2	NIPs	1.03 ± 0.61 µM
gG2	Free anti-HSV-2 Fab	19.7 ± 0.41 nM

## Data Availability

The data produced and analyzed in this study are not publicly accessible. However, interested parties may obtain the data by making a reasonable request to the corresponding authors.

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
