# Peer review of "Functionalized PLGA-Based Nanoparticles with Anti-HSV-2 Human Monoclonal Antibody: A Proof of Concept for Early Diagnosis and Targeted Therapy"

_pharmaceutics, 2024, doi:10.3390/pharmaceutics16091218_

Round 1

Reviewer 1 Report

Comments and Suggestions for Authors

Manuscript title: Development of Nano-Immuno-Probes as a Proof of Concept for Highly Sensitive Early Diagnosis and Targeted Therapy

Manuscript ID: pharmaceutics-3134558

The authors reported a biotechnological platform for selectively targeting viral infections caused by Human Herpes Simplex Virus, using PLGA-based polymeric nanocarriers coupled to recombinant Fab antibody wreckages, able to explicitly bind to viral glycoprotein G2. The proposed review is fascinating and well-organized. However, this lacks originality and creativity in gathering and segregating more information on this hot topic. I would recommend sticking to some core field of research discussions. This virus is therefore the paradigm of the potential pathogen that requires a specific and selective approach to early diagnose and handle these delicate patients. The manuscript can be further improved and enhanced by adding a few more insightful and comprehensive information on this interesting topic.

Below are the major comments:

1)      The title should be rephrased in a more meaningful and dwarf way.

2)      Truncate the abstract and provide the key information on the proposed article with critical parameters LOD, sensitivity, dynamic range, and reproducibility.

3)      What is the novelty and significance of this proposed review? How is this different from several existing similar state-of-the-art?

4)      The introduction should be a more comprehensive and detailed discussion of the fundamentals of state-of-the-art needs including the role of drug delivery, different pathogens, and target therapeutics.

5)      Is it reliable on biological infectious samples for targeted delivery? What is the impact of this proposed work in the pharmaceutical industry?

6)      Authors should consider adding new recently reported relevant articles to strengthen the proposed work: doi.org/10.1016/j.sna.2023.114385; 10.1186/s12943-023-01865-0

7)      What is the role of Nano-Immuno-Probes synthesis? How temperature control is significant? Characterization technique? Why especially PLGA-PEG-Mono-Sulfone NPs?

8)      Section 3 should be “Results and Discussion”

9)      How PCR technique is efficiently used and why gel electrophoresis? Why not fluorescence?

10)  Label the ladder in Fig 2 and 6

11)  Fig 3 resolution is poor and also provide the X-axis title. What are the values 90, 128, 257, 331, and so on.

12)   Fig 4 doesn’t provide much information or it's unclear.

13)  Fig 5 adds the magnification rate and SEM images are not clear.

14)  Add the Y-axis title in Fig 7(b)

15)  Materials and methods should be more specifically discussed and an additional section needs to be added.

16)  Authors should provide a summary comparison table on recently reported literature in self-heating elements with LOD, sensitivity, reproducibility, and application.

17)  What are the challenges and limitations of this approach?

18)  Add the appendix with all the abbreviations

19)  All the references should be rephrased and rechecked.

20)  Continuity flow is missing in most of the paragraphs.

21)  Improve the English language and Grammatical errors are too many.

Comments on the Quality of English Language

Attached

Author Response

1)      The title should be rephrased in a more meaningful and dwarf way.
Response: we thank the reviewer for the suggestion. The title has been rephrased to better reflect the content of the work and provide a more meaningful understanding.

2)      Truncate the abstract and provide the key information on the proposed article with critical parameters LOD, sensitivity, dynamic range, and reproducibility.
Response:  the abstract has been revised accordingly.

3)      What is the novelty and significance of this proposed review? How is this different from several existing similar state-of-the-art?

Response: We thank the referee for giving us the opportunity to clarify this topic. First of all, the manuscript is not a review but an experimental work; nevertheless, to further clarify its novelty, we have added information to the introduction that more clearly and comprehensively highlights how this work differs from existing literature, lines 69-79.

4)      The introduction should be a more comprehensive and detailed discussion of the fundamentals of state-of-the-art needs including the role of drug delivery, different pathogens, and target therapeutics.
Response:  We followed the reviewer’s comment by revising the introduction in lines 47-52.

5)      Is it reliable on biological infectious samples for targeted delivery? What is the impact of this proposed work in the pharmaceutical industry?
Response: The effective development and application of these nanodevices could significantly boost the pharmaceutical market for advanced drug delivery systems. We added this information in the “Conclusion” section, lines 458-461.

6)      Authors should consider adding new recently reported relevant articles to strengthen the proposed work: doi.org/10.1016/j.sna.2023.114385; 10.1186/s12943-023-01865-0
Response: Thank you for your suggestions. The article 10.1186/s12943-023-01865-0 has been added to the reference list.

7)      What is the role of Nano-Immuno-Probes synthesis? How temperature control is significant? Characterization technique? Why especially PLGA-PEG-Mono-Sulfone NPs?
Response: We appreciate the referee for giving us the opportunity to clarify this matter. The requested information has been included in the relevant sections: "Introduction" and "Conclusion”, lines 86-92 and 455-456.

8)      Section 3 should be “Results and Discussion”
Response: We thank the reviewer for the suggestion. Though, we think it would be better to follow the 'instructions for authors', according to which the manuscript sections should be Introduction, Materials and Methods, Results, Discussion, and Conclusions.

9 How PCR technique is efficiently used and why gel electrophoresis? Why not fluorescence?
Response: The PCR was essential for adding a Histidine tag to the C-terminal end of the heavy chain (HC) of the anti-HSV-2 Fab. This step was crucial for NPs functionalization and NIPs synthesis. The technique was efficiently used due to the primers employed and the optimization of PCR conditions, as described in section S1.1 Light and Heavy Chains Cloning of the supporting information. After the amplification reaction, we used gel electrophoresis to separate and purify the PCR products of interest, allowing us to proceed with cloning and obtain the pComb3/TIG-LC-HCHIS vector encoding the Histidine tagged anti-HSV-2 Fab (see S1.1 of the Supporting Information).

10)  Label the ladder in Fig 2 and 6
Response: The labeling of the ladder in Fig 2 and Fig 6 (now Fig 3 and 5, respectively) has been done.

11)  Fig 3 resolution is poor and also provide the X-axis title. What are the values 90, 128, 257, 331, and so on.
Response: Figure 3 (now Figure S1) was corrected. NTA analysis provided us the particle size distribution profile and concentrations measurements; each peak corresponds to statistical relevant subpopulation with a specific diameter, therefore numbers in Figure indicate the size values (nm) of the nanoparticles corresponding to the peaks.

12)   Fig 4 doesn’t provide much information or it's unclear.
Response: The figure has been removed and replaced with Table 2.

13)  Fig 5 adds the magnification rate and SEM images are not clear.
Response: The magnification rate (100K) has been added to Fig. 5 (now Fig 4). We have included an inset in Figure 5c (now Fig 4c) to clarify TEM images.

14)  Add the Y-axis title in Fig 7(b)
Response: We apologize, the Y-axis title was added to Figure 7b (now Figure 6b).

15)  Materials and methods should be more specifically discussed and an additional section needs to be added.

Response: We believe that the Materials and methods section is thoroughly discussed. However, we revised sentences in the following lines: lines 113-115, 125-127, 146- 147, 155-162, 225-229, 242-243; at line 133 we added the concentration of the antibiotics. A “supporting information” file has already been provided.

16)  Authors should provide a summary comparison table on recently reported literature in self-heating elements with LOD, sensitivity, reproducibility, and application.
Response: We thank the referee for the suggestion, but this work is focused on the results of the novel platform we have assembled for the selective delivery of active compounds to target HSV-2 and we prefer to strengthen and comment our results.

17)  What are the challenges and limitations of this approach?
Response: The approach described, while promising, does face several challenges and limitations. We have added some to the conclusion section, lines 456-457.

18)  Add the appendix with all the abbreviations
Response: the appendix was added.

19)  All the references should be rephrased and rechecked.
Response: All the references have been rephrased and rechecked.

20)  Continuity flow is missing in most of the paragraphs.
Response: Continuity flow has been improved throughout all paragraphs to enhance readability.

21)  Improve the English language and Grammatical errors are too many.
Response: Grammatical errors have been corrected, and the English language has been improved to make the manuscript more readable and understandable.

Reviewer 2 Report

Comments and Suggestions for Authors

The work reported by Mariotti et al titled "Development of Nano-Immuno-Probes as a Proof of Concept 2 for Highly Sensitive Early Diagnosis and Targeted Therapy " reports the synthesis of a fab targeting the glycoprotein of the HSV-2 virus and its conjugation on polymeric (PLGA) particles achieving a short of a theragnostic. Whereas the synthesis of the fab is properly described and also the affinity for the targetted G2 studied by SPR has been critically discussed, the employment of the nanoparticle is weakly supported and not properly investigated. As briefly (too Briefly) discussed by the authors in the discussion, the use of NP enables the diagnosis and treatment of the virus but the PLGA particles here reported can be used neither for diagnosis nor treatment since they are not labeled and not loaded with an antiviral drug. As discussed by the authors, SPR data showed that free fab is more efficient in recognizing G2 than NIPs.

I suggest authors reorganize the manuscript bringing the focus to the fab synthesis and characterization. The functionalization of particles could be moved to the end of the manuscript as a proof-of-concept of the possibility of conjugating it to nanomaterials for further applications.

Here are specific comments on the nanoparticles-related data presented:

·       A scheme summarizing the synthesis of the NIP would help the reader to better understand the procedure. As described, it is not clear if PLGA-PEG-NH2 and PLGA-PEG-Mono-Sulfone were first prepared as polymers and then precipitated as nanoparticles or if particles were already obtained with PLGA-PEG-NH2 co-polymer and subsequently functionalized. DLS data are meaningful only for particles and not for polymers.

·       Although the measurement is accomplished with the same instrument (Zetasizer), the type of analysis used for the analysis of the particle size (Dynamic light scattering) is different than the one used for measuring the zeta potential, which is called Electrophoresis light scattering (ELS). The principle is the same but the particle movement is not related to Bowian emotion (DLS) but to the applied electric field (es https://www.webofscience.com/wos/woscc/full-record/WOS:000535234000105, https://doi.org/10.1021/acsomega.9b00965, ). Usually, the ELS results are reported in tables since the graphs exported from the software don't have relevant additional information to the values given by the software. I suggest the authors report the values in a table as they did for the size and remove fig4

·       I would also move the NTA analysis to the supporting information since it does not add much information. More information should be added to the methods since it is not easy to analyze unlabeled polymeric particles with such techniques…

·       Regarding the TEM analysis:

o   should be specified that it is only with low-contrast materials (such as PLGA) that contrast agents are needed for the analysis.

o   If the bar scale is the same between the images reported in fig5, then there is a big variation of the particle size between a-b and c-d. Why is that?

o   It is not clear what the control reported in d represents and how the authors confirmed that the antibody recognition occurred by TEM. Gold particles should be visible by TEM and are usually also negatively charged. The appropriate methods to confirm the conjugation efficacy of the NIPs for the functionalized gold particles would be 1) Zetasizer analysis since both size and the surface charge will change, and 2) UV-Vis analysis thanks to the absorption of gold nanoparticles around 400 nm. NP with the gold particles should be used as the control sample also in the SPR validation experiment.

Author Response

I suggest authors reorganize the manuscript bringing the focus to the fab synthesis and characterization. The functionalization of particles could be moved to the end of the manuscript as a proof-of-concept of the possibility of conjugating it to nanomaterials for further applications.

Here are specific comments on the nanoparticles-related data presented:

  • A scheme summarizing the synthesis of the NIP would help the reader to better understand the procedure. As described, it is not clear if PLGA-PEG-NH2 and PLGA-PEG-Mono-Sulfone were first prepared as polymers and then precipitated as nanoparticles or if particles were already obtained with PLGA-PEG-NH2 co-polymer and subsequently functionalized. DLS data are meaningful only for particles and not for polymers.

Response: We thank the referee for the suggestion, and to enhance understanding, we have added a figure to the manuscript (see Figure 1).

  • Although the measurement is accomplished with the same instrument (Zetasizer), the type of analysis used for the analysis of the particle size (Dynamic light scattering) is different than the one used for measuring the zeta potential, which is called Electrophoresis light scattering (ELS). The principle is the same but the particle movement is not related to Bowian emotion (DLS) but to the applied electric field (es https://www.webofscience.com/wos/woscc/full-record/WOS:000535234000105, https://doi.org/10.1021/acsomega.9b00965, ). Usually, the ELS results are reported in tables since the graphs exported from the software don't have relevant additional information to the values given by the software. I suggest the authors report the values in a table as they did for the size and remove fig4

Response: Figure 4 has been removed and replaced by a new table, which is now Table 2 in the manuscript.

  • I would also move the NTA analysis to the supporting information since it does not add much information. More information should be added to the methods since it is not easy to analyze unlabeled polymeric particles with such techniques…

    Response. The NTA data has been moved to the supplementary materials.

  • Regarding the TEM analysis:

o   should be specified that it is only with low-contrast materials (such as PLGA) that contrast agents are needed for the analysis.

Response: Revised accordingly.

o   If the bar scale is the same between the images reported in fig5, then there is a big variation of the particle size between a-b and c-d. Why is that?

Response: Yes, the NPs population has different size, as shown by DLSA and NTA data. We have included an inset in Figure 5c (now Fig 4c) to highlight the presence of larger NIPs.

o   It is not clear what the control reported in d represents and how the authors confirmed that the antibody recognition occurred by TEM. Gold particles should be visible by TEM and are usually also negatively charged. The appropriate methods to confirm the conjugation efficacy of the NIPs for the functionalized gold particles would be 1) Zetasizer analysis since both size and the surface charge will change, and 2) UV-Vis analysis thanks to the absorption of gold nanoparticles around 400 nm. NP with the gold particles should be used as the control sample also in the SPR validation experiment.

Response. We thank the referee for the suggestion. The figure 5d represents the negative control obtained by omitting the primary antibody in the IEM reaction. Negative control was obtained by substituting PBS to the anti-human Fab antibody (primary antibody), as described in the methods. Indeed, we have characterized using Zetasizer both NPs and NIPs. Specifically, our results are reported in Table 1 and Table 2 and show that the NIPs are larger with respect to the NPs not functionalized, as well as the Z-potential is strongly modified by the external moiety due to Fab conjugation. TEM analysis was a final step to visualize this difference. The IEM technique is highly specific as it leverages the binding between antigen and antibody. The presence of colloidal gold particles (electron-dense) exclusively on the surface of the NIPs after incubation with the primary antibody, and their absence in the negative control (see above), demonstrate the specificity of the antigen (Fab) and antibody binding. Therefore, the colloidal gold particles allow for the detection of the presence of Fab on the NIPs.

Reviewer 3 Report

Comments and Suggestions for Authors

The manuscript "Development of Nano-Immuno-Probes as a Proof of Concept 2 for Highly Sensitive Early Diagnosis and Targeted Therapy" by Mariotti et al. describes the synthesis, functionalization, characterization of nanoparticles for Human Herpes Simplex Virus 2 gG2 protein identification as sensor. The manuscript is clearly written, methods are well described, and conclusions supported by results. For this reason I recommend this manuscript for pubblication. 

To be note some minor corrections:

-gG2 protein abbreviature should be inserted the first time that glycoprotein is cited

-in the SPR method description the buffer immobilizzation employed is not specify, please add the full conditions of the immobilizzation. Are all solutions for binding and kinetic experiments performed in PBS?

Author Response

The manuscript "Development of Nano-Immuno-Probes as a Proof of Concept 2 for Highly Sensitive Early Diagnosis and Targeted Therapy" by Mariotti et al. describes the synthesis, functionalization, characterization of nanoparticles for Human Herpes Simplex Virus 2 gG2 protein identification as sensor. The manuscript is clearly written, methods are well described, and conclusions supported by results. For this reason I recommend this manuscript for pubblication. 

Response: Thank you.

To be note some minor corrections:

-gG2 protein abbreviature should be inserted the first time that glycoprotein is cited
Response: Revised accordingly.

-in the SPR method description the buffer immobilizzation employed is not specify, please add the full conditions of the immobilizzation. Are all solutions for binding and kinetic experiments performed in PBS?

Response: We thank the referee for the suggestion and believe that the Materials and Methods related to SPR are described in sufficient detail. However, we have added some information regarding the immobilization process in the “Materials and Methods” section. Both NIPs and the antibody were dissolved in PBS1X for the binding analyses.

Round 2

Reviewer 1 Report

Comments and Suggestions for Authors
The authors have made most of the changes suggested to the revised version based on comments. However, the Authors have forgotten to add this relevant reference doi.org/10.1016/j.sna.2023.114385 to the introduction section to support the literature. Also, please add the SWOT analysis for this proposed work. Overall, seems to be good fascinating work. 
Comments on the Quality of English Language
The authors have made most of the changes suggested to the revised version based on comments. However, the Authors have forgotten to add this relevant reference doi.org/10.1016/j.sna.2023.114385 to the introduction section to support the literature. Also, please add the SWOT analysis for this proposed work. Overall, seems to be good fascinating work. 

Author Response

The authors have made most of the changes suggested to the revised version based on comments. However, the Authors have forgotten to add this relevant reference doi.org/10.1016/j.sna.2023.114385 to the introduction section to support the literature. Also, please add the SWOT analysis for this proposed work. Overall, seems to be good fascinating work.

We thank the referee for the suggestion. We added the reference and introduced a SWOT analysis as a supplementary table.

Reviewer 2 Report

Comments and Suggestions for Authors

No further comments.

Author Response

We thank the referee for the support.